# Improving Image Quality Assessment Based on the Combination of the Power Spectrum of Fingerprint Images and Prewitt Filter

Ting-Wei Shen [1] , Ching-Chuan Li [2], Wan-Fu Lin [2], Yu-Hao Tseng [2], Wen-Fang Wu [1], Sean Wu [3,*],
Zong-Liang Tseng [2,*] and Mao-Hsiu Hsu [4,*]

1 Department of Mechanical Engineering, National Taiwan University, Taipei 106, Taiwan;
  d05543008@ntu.edu.tw (T.-W.S.); wfwu@ntu.edu.tw (W.-F.W.)
2 Department of Electronic Engineering, Ming Chi University of Technology, New Taipei 243, Taiwan;
  u07157008@mail2.mcut.edu.tw (C.-C.L.); u07157010@mail2.mcut.edu.tw (W.-F.L.);
  u07157031@mail2.mcut.edu.tw (Y.-H.T.)
3 Department of Chemical and Materials Engineering, Lunghwa University of Science and Technology,
  Taoyuan 333, Taiwan
4 FocalTech Systems Co., Ltd., Hsinchu 300, Taiwan
* Correspondence: wusean.tw@gmail.com (S.W.); zltseng@mail.mcut.edu.tw (Z.-L.T.);
  mh.hsu@focaltech-electronics.com (M.-H.H.)

**Abstract:** The assessment of fingerprint image quality is critical for most fingerprint applications. It has an impact on the performance and compatibility of fingerprint recognition, authentication, and built-in cryptosystems. This paper developed an improved fingerprint image quality assessment derived from the image power spectrum approach and combined it with the Prewitt filter and an improved weighting method. The conventional image power spectrum approach and our proposed approach were implemented for accuracy and reliability tests using good, faulty, and blurred fingerprint images. The experimental results showed the proposed algorithm accurately identified the sharpness of fingerprint images and improved the average difference in FIQMs to 61% between three different levels of blurred fingerprints compared with that achieved by a conventional algorithm.

**Keywords:** fingerprint; image quality assessment; image power spectrum; Prewitt filter; image processing

## 1. Introduction

Fingerprints are widely recognized as one of the most commonly utilized and acceptable features for personal identification [1–4]. This is explained by two reasons: firstly, fingerprints are one-of-a-kind and permanent in a person's life, making them ideal long-term indicators of human identity [5,6]; secondly, fingerprints have one of the highest degrees of dependability among all biometric markers [7,8], and forensic specialists have employed them extensively in criminal investigations [9]. Fingerprint technologies are rapidly being used for user authentication in a variety of civilian and business applications [10].

Most fingerprint matching methods use global or local fingerprint features, and feature extraction is highly affected by the integrity and quality of fingerprint images [11,12]. Fingerprint features include ridge and valley clarity, as well as minutiae, core, and delta points [13,14]. Although the classic power spectrum method is frequently used for fingerprint image quality assessment [15], an algorithm that precisely and rapidly assesses the subjective quality of a fingerprint image still is an important and ongoing objective [16].

Fingerprint Image Quality Assessment (FIQA) has attracted efforts from both academic and industrial areas. The existing studies can be classified into three main types of approaches [17–19], namely, (a) single feature approaches, (b) segmentation approaches, and (c) multi-feature approaches.

Single feature approaches are generally carried out in two ways: global methods and local methods. To assess the quality, global methods examine the entire fingerprint image, whereas local methods extract features from each block and assess quality from those features. Lee et al. [20] proposed an approach based on the Fourier spectrum of the fingerprint image as well as reviewing three approaches based on local standard deviation [21], directional contrast of local block [22], and the Gabor feature [23].

Segmentation approaches are based on segmenting the directional block and non-directional block with a threshold [24]. The fingerprint image quality matrices are represented as a ratio of the area of directional blocks to the image. However, the threshold of directional blocks may differ for different sensors. Shen et al. [23] proposed a technique that employs 8-directional Gabor filters to segment the foreground from the image.

Multi-feature approaches combine different fingerprint image quality assessment approaches and classify the blocks into variant levels with thresholds. NIST fingerprint image quality (NFIQ) was proposed by Tabassi et al. [25]. This approach classifies results into five levels via a trained model of a neural network and estimates a matching score utilizing 11 dimensional features including ridge orientation flow, local ridge curvature, and local contrast.

Although the abovementioned approaches show a significant FIQA improvement, the complicated redesign has a higher cost and is time-consuming. For instance, learning-based methods heavily rely on data-driven techniques. Namely, they require a massive amount of labeled data and utilize pretrained networks [26]. This is time-consuming and requires additional computer power. Therefore, a modification of the conventional algorithm (i.e., Power Spectrum) would enable an efficient development to improve FIQA.

In this study, we developed a method for quantitatively and objectively assessing image quality derived from the classic Power Spectrum method, which is frequently used for fingerprint image quality [15]. Its main advantage is that it eliminates reliance on specific calibration images, such as Slanted-edge, Siemens star, and sinusoidal pattern (Figure 1). It is unnecessary to capture the image-specific feature targets, such as contour lines and edges as well as corners [15]. The proposed assessment method differs from the conventional Power Spectrum approaches in that it applies the Prewitt high-pass filter to fingerprint images before carrying out Fourier transformation and runs an improved weighting method, taking the frequency as weighted coefficients, to improve evaluation accuracy.

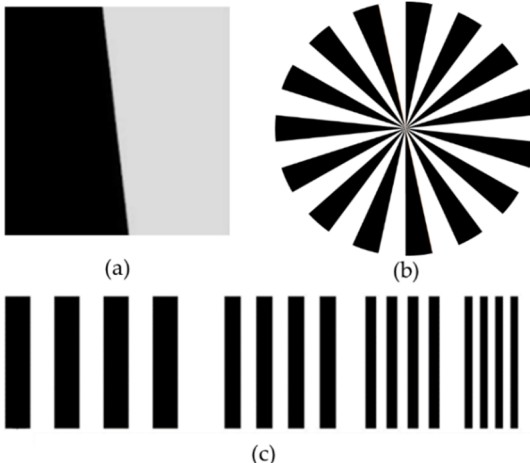

**Figure 1.** (**a**) Slanted-edge pattern, (**b**) Siemens star pattern, and (**c**) Sinusoidal pattern.

## 2. Methods

### 2.1. Development of FIQA Based on the Power Spectrum

In this paper, we focus discussion on the Power Spectrum method, and the conventional Power Spectrum algorithm is implemented and compared with our proposed approach. This section explains how the proposed FIQA method operates. The calculation

steps are shown in Figure 2, and a detailed description of each step is provided in the succeeding sections.

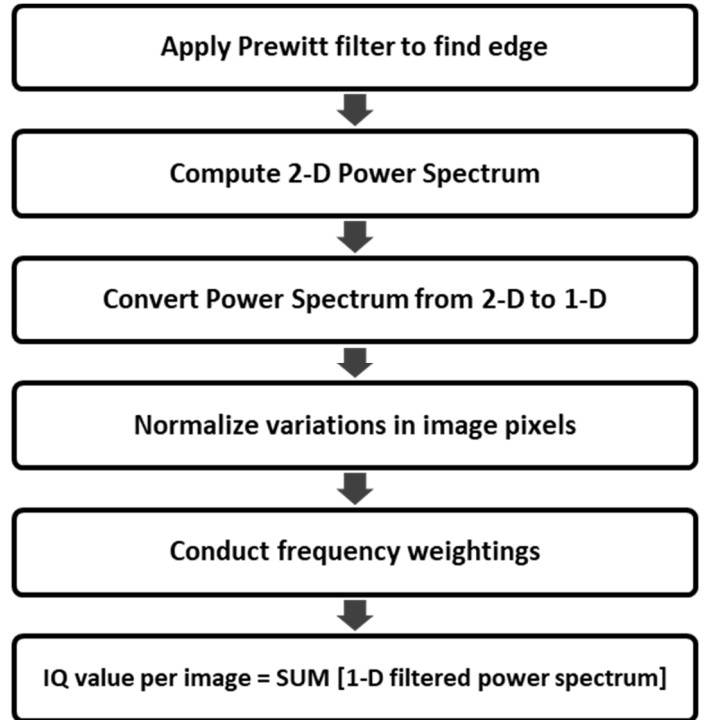

**Figure 2.** The Proposed FIQA process.

### 2.2. The Prewitt Edge Method

The Prewitt edge method is a high-pass image filtering approach that can emphasize high-spatial-frequency regions that correspond to edges. Typically, it is used to find the approximate absolute gradient magnitude at each point in an input grayscale image.

The Prewitt operator consists of a pair of 3 × 3 convolution kernels (Figure 3) [27]. These kernels are designed to respond maximally to edges running vertically and horizontally relative to a pixel grid. The kernels can be applied separately to an input image to produce separate measurements of the gradient component in each orientation (denoted as $G_x$ and $G_y$). These measurements can then be combined to determine the absolute magnitude of the gradient at each point and the orientation of that gradient. The gradient magnitude is given by Equation (1). Before calculating the image power spectrum of fingerprints, we applied the Prewitt high-pass filter to fingerprint images.

$$h(x,y) = G = \sqrt[2]{G_x{}^2 + G_y{}^2} \tag{1}$$

| 1 | 1 | 1 |
|---|---|---|
| 0 | 0 | 0 |
| −1 | −1 | −1 |

| −1 | 0 | 1 |
|---|---|---|
| −1 | 0 | 1 |
| −1 | 0 | 1 |

**Figure 3.** Prewitt convolution kernel.

### 2.3. Calculation of the 2-D Image Power Spectrum

Historically, many methods of calculating an image power spectrum have been developed. For instance, spatial frequency response (SFR) based on ISO 12233 is widely used in the field of image quality assessment. It provides a complete profile of the power spectrum (spatial frequency) of images. In that document (ISO 12233), a test chart such as a Slanted-edge or Siemens star pattern, is employed to estimate the spatial resolving capability. However, this method no longer meets the demand in some cases. For example, to achieve passive focus function for natural scenes, or specific targets, such as fingerprints. In these cases, it is not possible to take designed targets or a constant scene; thus, it cannot obtain the fingerprint image quality by the traditional SFR method.

Contributed to by modern computer performance, in this study, we determine the power spectrum of images calculating its 2D fast Fourier transformation (FFT). Given an image with $M \times M$ pixels, $h(x, y)$ is the gray level, with rectangular spatial coordinates $x$ and $y$ ranging from 0 to $M - 1$, and $H(u, v)$ is the corresponding discrete Fourier transformation. The discrete Fourier transformation of the image is defined as Equation (2).

$$H(u,v) = \sum_{x=0}^{M-1} \sum_{y=0}^{M-1} exp\left[-2\pi iy\frac{v}{M}\right] exp\left[-2\pi ix\frac{u}{M}\right] h(x,y), \ u,v = -\frac{M}{2}\ldots\frac{M}{2}. \quad (2)$$

To relate the orthogonal $u$ and $v$ indices of $H(u, v)$ to spatial frequency components of the input image, these indices must be normalized based on the number of pixels, $M$, in the $x$ and $y$ directions. This process results in $u/M$ and $v/M$ in units of cycles per pixel width. We can acquire an estimate of the power spectrum by multiplying the Fourier terms $H(u, v)$ by their complex conjugate, that is, the absolute square of $H(u, v)$ is defined as Equation (3) [28].

$$|H(u,\ v)|^2 = H\overline{H} = (a+bi)(a-bi) = a^2 + b^2. \quad (3)$$

### 2.4. Conversion of the 2-D Image Power Spectrum to 1-D

Because of the large number of data points in the 2-D power spectrum, comparing images quality is difficult in 2-D power spectrum. Converting the 2-D power spectrum into a 1-D form (Figure 4) is necessary, which enables more tractable image comparisons. We generated the 1-D power spectrum by averaging the power spectrum contained within bands of frequencies, where the frequency is defined as Equation (4), from the origin, i.e., the center of the power spectrum, to the power value at $(u, v)$, in units of cycles per pixel.

$$\rho = \sqrt{u^2 + v^2}/M = Cycle/Pixel \quad (4)$$

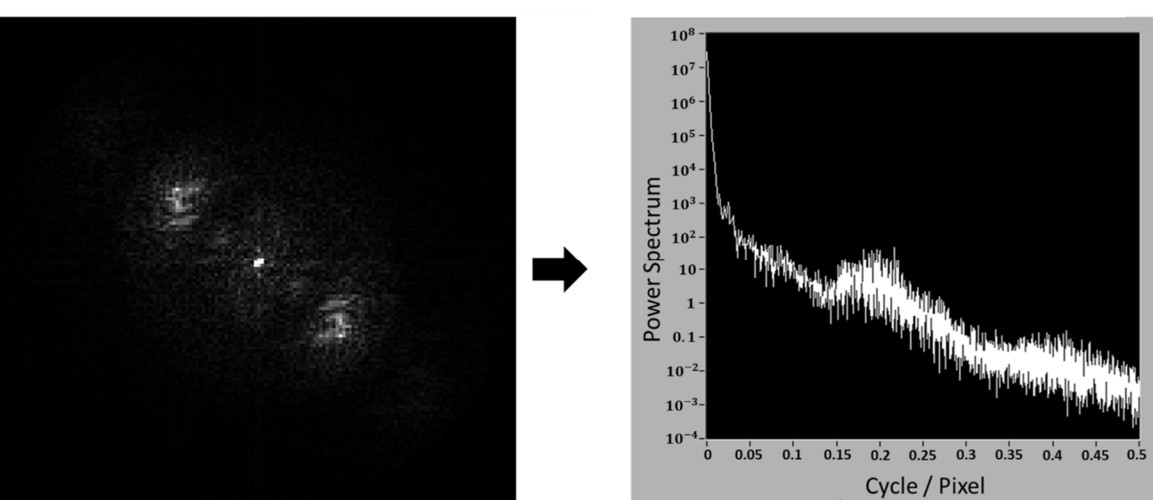

**Figure 4.** Conversion of the 2-D Image Power Spectrum to 1-D.

*2.5. Normalization of Image Pixel Size*

The value of the image power spectrum is also proportional to the number of pixels in the image, and all the finger images are square in this article. Thus, eliminating the influence of image pixel size requires normalizing variations in image pixel size by dividing the 2-D power spectrum by the total number of pixels in the image ($M^2$).

*2.6. Frequency Weightings of the Image Power Spectrum*

Furthermore, the influence of frequency should be considered becausethe high-frequency contents of an image correspond to image details. Accordingly, the image's power spectrum and the corresponding frequency are multiplied as a weighting calculation, enabling a more accurate image quality evaluation. The combination of frequency weightings and pixel size normalization generates a normalized 2-D power spectrum $P(u, v)$, obtained using Equation (5).

$$P(u,v) = \rho \cdot |H(u, v)|^2 / M^2 \cdot \tag{5}$$

Given that polar coordinates were used in the remaining calculations, Equation (5) becomes Equation (6).

$$P(\rho, \theta) = \rho \cdot |H(\rho, \theta)|^2 / M^2, \ \ \theta = tan^{-1} v / u \ \ \cdot \tag{6}$$

*2.7. Fingerprint Image Quality Metrics*

Fingerprint Image quality metrics (FIQMs) are derived from the double summation of the pixel size-normalized 2-D image power spectrum $P(\rho, \theta)$ weighted by frequency. We can obtain a single value, FIQMs, from a 2-D image power spectrum with an equation that represents image sharpness. A FIQMs is determined using Equation (7).

$$FIQMs = \frac{1}{M^2} \sum_{\theta=0^\circ}^{360^\circ} \sum_{\rho=0.01}^{0.5} \rho \cdot |H(u, v)|^2 = \sum_{\theta=0^\circ}^{360^\circ} \sum_{\rho=0.01}^{0.5} P(u,v). \tag{7}$$

## 3. Experiments and Results

Two sets of experiments were conducted to verify the correctness of the algorithm put forward in this work. In the first set of experiments, 150 fingerprint images were selected from a database and blurred to three different levels ($3 \times 3$, $5 \times 5$, and $7 \times 7$ kernels) using Gaussian smoothing (Figure 5). Then, 150 sets of images (450 images) were imported into the proposed Power Spectrum + Prewitt filter algorithm, the conventional Power Spectrum algorithm and the Structural Similarity Index (SSIM) to determine whether the evaluation values of the blurred images were negatively correlated with the degree of blurring. SSIM is a classic full reference metric, which measures the degradation of image quality due to data compression or transmission losses. In the second set of experiments, 150 good and faulty fingerprint images (Figure 6) were selected from a database to test whether the proposed and conventional algorithms could correctly classify the two groups. Finally, a comparison of the experimental results was performed between the proposed algorithm, the Power Spectrum algorithm, and SSIM. All the algorithms were implemented using LabVIEW.

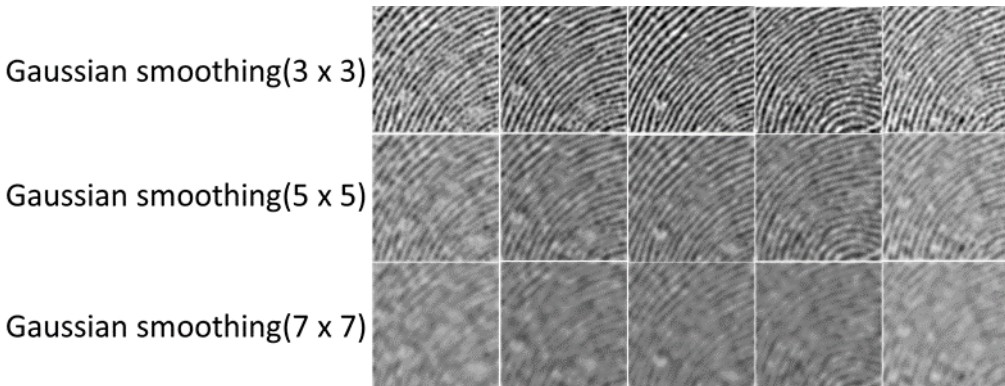

**Figure 5.** Fingerprint images with three different levels of blurring.

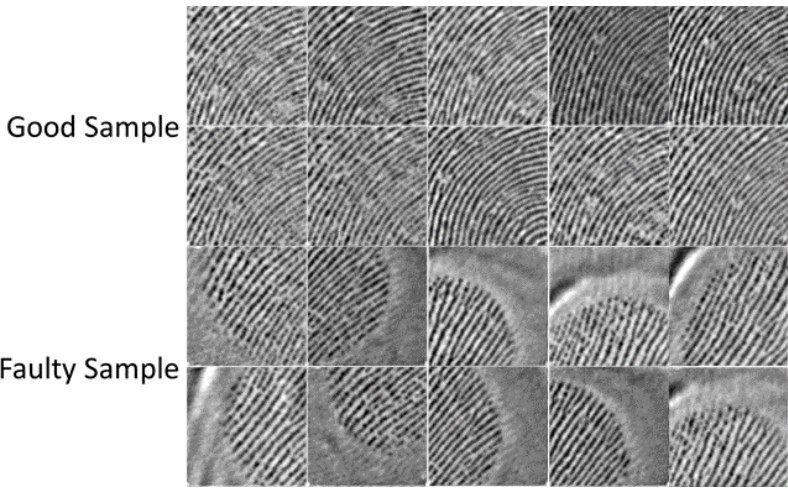

**Figure 6.** Good and faulty fingerprint images.

### 3.1. Database

In this work, we used our own fingerprint data, which were captured with under OLED panel display image sensor, and the dataset was composed of 2520 images (126 fingers with 20 samples per finger). The resolution of each sample is 813 DPI and 192 × 192 pixels.

### 3.2. Experiment 1: Sharpness Assessment

The fingerprint images blurred to different degrees were entered into the Power Spectrum + Prewitt Filter algorithm developed in this research, the conventional Power Spectrum algorithm, and SSIM. The FIQMs were recorded as a box and whisker plot and line chart (Figure 7). As can be seen from the graphs, the FIQMs of the Power Spectrum + Prewitt Filter approach decreased as the degree of blurriness increased in severity, and this trend was observed in all the sample images. The findings of the sharpness experiments in the conventional algorithm showed similar results to the proposed algorithm, which identified the sharpness of fingerprint images correctly as well; however, the plots indicate that the results of the conventional algorithm had a smaller interval between each kernel group, which means the proposed algorithm could increase the average difference in FIQMs from 0.124 to 0.20 among the 3 × 3, 5 × 5, and 7 × 7 kernels via Gaussian smoothing. The results indicated that SSIM had good performance in sharpness assessment; namely, the FIQMs of different kernel set (3 × 3, 5 × 5, and 7 × 7) had clear gaps between each other, and the central tendency of the FIQMs was better than other algorithms. However, SSIM required reference images; for instance, every test sample needed its origin distortionless image for reference; hence, it is not a practical candidate for FIQA.

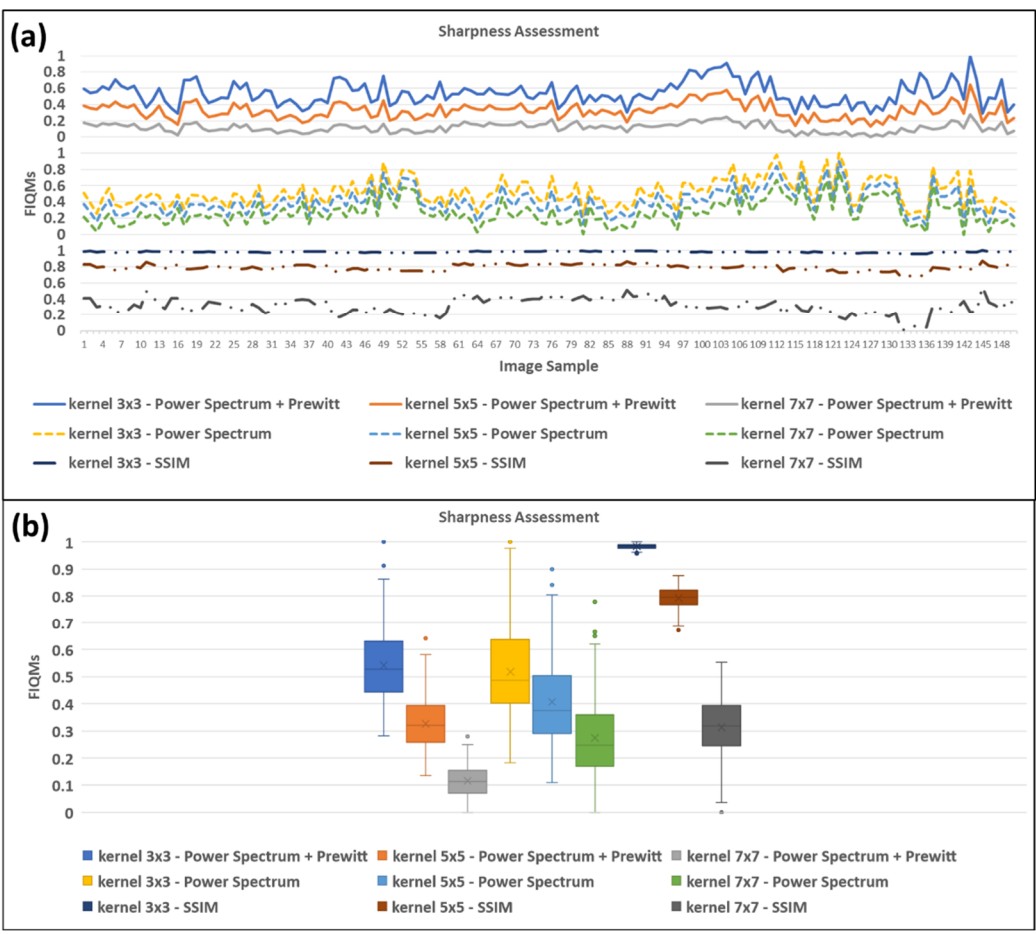

**Figure 7.** Experimental results of Sharpness Assessment: (**a**) box and whisker plot and (**b**) line chart.

### 3.3. Experiment 2: Good/Faulty Assessment

The good and faulty fingerprint image samples were entered into the proposed Power Spectrum + Prewitt Filter algorithm, the conventional Power Spectrum algorithm, and SSIM, after which the results were recorded. In the experimental results of the proposed algorithm, the box and whisker plot and line chart (Figure 8) showed that the results of the Power Spectrum + Prewitt algorithm had a clear difference between the two groups of data, and this finding was consistent with the trend wherein the FIQMs of the good samples were higher than those of faulty samples. On the other hand, the conventional algorithm could not identify the quality of fingerprint images; with the results even good fingerprints had lower FIQMs. With regard to SSIM, the charts reveal that it could not distinguish good samples from faulty samples, when we used a good fingerprint sample for reference. For comparison, the differences in FIQMs and positive/negative aspects between the proposed and conventional algorithm accuracy are summarized in Tables 1 and 2.

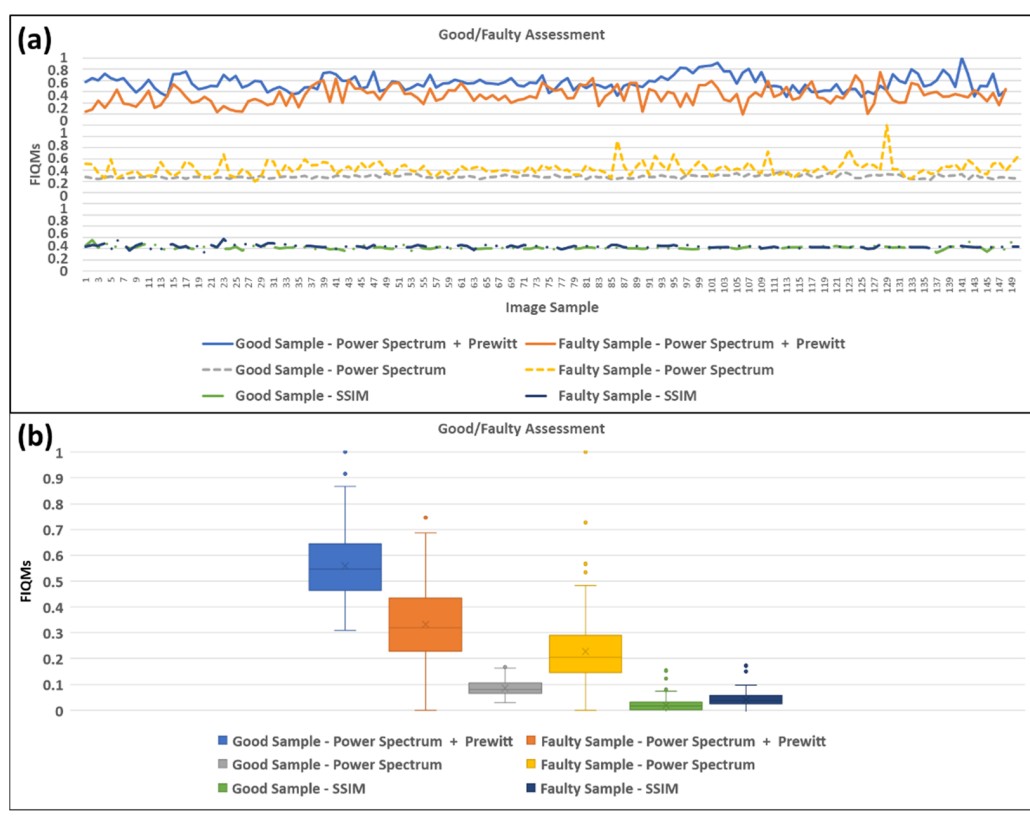

**Figure 8.** Experimental results of Good/Faulty Assessment: (**a**) box and whisker plot and (**b**) line chart.

**Table 1.** FIQMs with different levels of blurring for the proposed and conventional algorithm.

| | Power Spectrum + Prewitt Filter | | | Power Spectrum | | | SSIM | | |
|---|---|---|---|---|---|---|---|---|---|
| Kernel | $3 \times 3$ | $5 \times 5$ | $7 \times 7$ | $3 \times 3$ | $5 \times 5$ | $7 \times 7$ | $3 \times 3$ | $5 \times 5$ | $7 \times 7$ |
| FIQMs(median) | 0.521 | 0.319 | 0.12 | 0.49 | 0.374 | 0.241 | 0.97 | 0.796 | 0.329 |
| Average Difference in FIQMs | 0.20 | | | 0.124 | | | 0.32 | | |

**Table 2.** Advantages and disadvantages of the proposed and conventional algorithm.

| | Power Spectrum + Prewitt Filter | Power Spectrum | SSIM |
|---|---|---|---|
| Advantage | Average difference in FIQMs increases up to 61%. | A classic approach commonly utilized in the FIQA field. | Good performance in sharpness assessment of FIQA. |
| Disadvantage | Higher computational demand. | Lower resolution in FIQMs and could not identify good or faulty samples. | Requires reference images and failed in good or faulty assessment. |

## 4. Conclusions

The proposed Power Spectrum + Prewitt filter FIQA could identify the sharpness of the fingerprint images and increased the average difference in FIQMs from 0.124 to 0.20, which is approximately a 61% improvement among the $3 \times 3$, $5 \times 5$, and $7 \times 7$ kernels via Gaussian smoothing. This value corresponds to an improvement in assessment resolution, and good and faulty samples were well identified; the plot shows a clear difference between the two groups of data, and the FIQMs of the good samples were higher than those of the faulty sample, as expected. The conventional Power Spectrum FIQA failed to distinguish between good and faulty fingerprints. Although SSIM had a good performance in sharpness assessment, it is not a suitable candidate for FIQA because it requires reference images; in

addition, it failed in the good/faulty assessment. The experimental results confirmed that the proposed FIQA outperformed the traditional approaches.

**Author Contributions:** Conceptualization, Z.-L.T. and M.-H.H.; formal analysis, T.-W.S., C.-C.L., W.-F.L. and Y.-H.T.; investigation, T.-W.S., C.-C.L., W.-F.L. and Y.-H.T.; resources, S.W. and Z.-L.T.; writing—original draft, T.-W.S. and W.-F.W. All authors have read and agreed to the published version of the manuscript.

**Funding:** This work was supported by the Ministry of Science and Technology, Taiwan, under Grant No. MOST 110-2221-E-131-028.

**Institutional Review Board Statement:** Not applicable.

**Informed Consent Statement:** Not applicable.

**Data Availability Statement:** Not applicable.

**Acknowledgments:** The authors would like to extend their heartfelt thanks for the instrument support at the Advanced Instrument Center of National Yunlin University of Science and Technology.

**Conflicts of Interest:** The authors declare no conflict of interest.

## Abbreviations

| | |
|---|---|
| FIQA | fingerprint image quality assessment |
| FIQMs | fingerprint image quality metrics |
| $G$ | gradient magnitude |
| $G_x$ | gradient component in the X-axis |
| $G_y$ | gradient component in the Y-axis |
| FFT | fast Fourier transformation |
| $M \times M$ | dimensions of image pixels |
| $h$ | the gray level |
| $x, y$ | rectangular spatial coordinates $x$, and $y$ ranging from 0 to $M - 1$ |
| $H$ | the value of corresponding discrete Fourier transformation |
| $u, v$ | rectangular spatial coordinates $u$ and $v$ ranging from $-M/2$ to $M/2$ |
| $\rho$ | frequency in units of cycles per pixel |
| $P$ | frequency weighting and pixel size normalized power spectrum |

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
