# Peer review of "Improving Image Quality Assessment Based on the Combination of the Power Spectrum of Fingerprint Images and Prewitt Filter"

_applsci, doi:10.3390/app12073320_

Round 1
Reviewer 1 Report
Overall the proposed method and presentation are ok, however, please look at the overall improvements claimed here (154%) of improvements. it should be (difference value / original value) i.e. ((0.23-0.149)/0.149) = 0.54 which indicates 54% of improvements rather than 154%.
Reviewer 2 Report
The manuscript is in much better shape than the previous version. However, there are some points in the manuscript that must be improved.
1.) The authors should mention that fingerprint image quality assessment (FIQA) is a smaller part of image quality assessment which is now dominated by deep learning. The authors could cite this paper for reference: No-reference image quality assessment with convolutional neural networks and decision fusion, 2022. It would be nice to write something about what restricts the use of machine learning in FIQA.
2.) The main contribution of the study is still rather unclear. The authors write that "Its main advantage is that it eliminates reliance on specific calibration images, such as sine wave diagrams and Siemens star diagrams." However, the experimental results are rather obscure in this respect. How does the proposed method eliminate the usage of sine wave diagrams? Moreover, it is not clear for readers who are not expert in FIQA what sine wave diagrams and Siemens star diagrams are.
3.) The acquisition of data is not clear. Do the authors use publicly available databases? Are benchmark databases available in the literature? Did the authors use own data? How was it acquired? Main characteristics (resolution, distortion types, etc.)? Could the authors publish more sample images of the applied database?
4.) A comparison to at least another state-of-the-art is missing from the paper.
Round 2
Reviewer 2 Report
I am satisfied with the authors' answers. I recommend this manuscript for publication.